# Nucleus Accumbens Chemogenetic Inhibition Suppresses Amphetamine-Induced Ultrasonic Vocalizations in Male and Female Rats

**DOI:** 10.3390/brainsci11101255

**Published:** 2021-09-22

**Authors:** Kate A. Lawson, Abigail Y. Flores, Rachael E. Hokenson, Christina M. Ruiz, Stephen V. Mahler

**Affiliations:** Department of Neurobiology & Behavior, University of California, Irvine. 1203 McGaugh Hall, Irvine, CA 92697, USA; lawsonk1@uci.edu (K.A.L.); abigaiyf@uci.edu (A.Y.F.); rhokenso@uci.edu (R.E.H.); cruiz1@uci.edu (C.M.R.)

**Keywords:** 50 kHz vocalizations, 22 kHz vocalizations, amphetamine, nucleus accumbens, chemogenetics, clozapine-n-oxide, UMAP, males, females

## Abstract

Adult rats emit ultrasonic vocalizations (USVs) related to their affective states, potentially providing information about their subjective experiences during behavioral neuroscience experiments. If so, USVs might provide an important link between invasive animal preclinical studies and human studies in which subjective states can be readily queried. Here, we induced USVs in male and female Long Evans rats using acute amphetamine (2 mg/kg), and asked how reversibly inhibiting nucleus accumbens neurons using designer receptors exclusively activated by designer drugs (DREADDs) impacts USV production. We analyzed USV characteristics using “Deepsqueak” software, and manually categorized detected calls into four previously defined subtypes. We found that systemic administration of the DREADD agonist clozapine-n-oxide, relative to vehicle in the same rats, suppressed the number of frequency-modulated and trill-containing USVs without impacting high frequency, unmodulated (flat) USVs, nor the small number of low-frequency USVs observed. Using chemogenetics, these results thus confirm that nucleus accumbens neurons are essential for production of amphetamine-induced frequency-modulated USVs. They also support the premise of further investigating the characteristics and subcategories of these calls as a window into the subjective effects of neural manipulations, with potential future clinical applications.

## 1. Introduction

Rats emit vocalizations in the ultrasonic range, but the fundamental function of these emissions is still controversial. These rat ultrasonic vocalizations (USVs) seem to be related to affective states experienced by the animal, and may also have socially communicative functions [1,2,3,4,5,6,7,8,9,10,11]. We are most interested in the potential usefulness of USVs for querying the subjective states of rats, such as those produced by addictive drugs, in behavioral neuroscience experiments. By carefully measuring and analyzing effects of neural manipulations on the quantity and quality of USVs produced, we hope to gain new information about how defined neural populations help generate rats’ affective states. Along with the intrinsic scientific importance of this question, we hope this approach may also help inform future development of brain intervention approaches for treating psychiatric disease in humans, including addiction.

Adult rats emit USVs in two main ranges that are thought to correspond roughly to negative and positive emotional or anticipatory states [2,4,12,13]: Lower-frequency USVs (LF; 18–30 kHz; also called “22 kHz” vocalizations) are linked to stress and fear states [14,15,16,17,18,19]; higher-frequency USVs (30–100 kHz) are linked instead to reward anticipation or experience [6,7,8,20,21,22,23]. LF vocalizations are generally not frequency-modulated, and can occur with short or long durations, potentially reflecting the intensity of negative effects [24,25,26]. In contrast, high-frequency USVs (also called “50 kHz” vocalizations), are more complex, with at least 14 distinct patterns [27]. They are emitted at a range of principal frequencies (30–100 kHz), durations, and combinations of acoustic elements. Of particular note, some vocalizations have a rapid oscillating component called “trills.” Other high-frequency USVs have a consistent principal frequency (Flat), and others have a variety of non-trill, but still frequency-modulated patterns (FM). Although many reports do not distinguish between these patterns of high-frequency USVs, there is some evidence that they are produced under distinct circumstances, and subserved by different neural substrates [28,29,30,31].

In order to determine whether potentially useful information is contained in rat USVs, and to determine the neural substrates responsible, USVs must first be accurately recorded and analyzed. Advances in microphone technology and tools for visualizing USVs allow for accurate analysis of their presence and characteristics, greatly facilitating this objective. However, this approach has been limited by the necessity of manually categorizing and quantifying calls, which is very time consuming. Therefore, several groups and companies have developed USV detection and analysis software in recent years [32,33,34,35,36,37]. Here, we employed one of these, a machine learning-driven detection and categorization tool called Deepsqueak [33], which detects rat USVs and extracts parameters of them with reasonable accuracy. We contend that optimizing this, or similar, approaches may hold the key to extracting rich information about rat subjective states from USV data, thus facilitating future studies of neural substrates in rats. 

In a step toward this aim, we here asked how reversibly inhibiting nucleus accumbens (NAc) neurons affects emission of rat USVs. NAc is a key anatomical substrate for high frequency USV production [22,38,39,40,41], and here, for the first time in the context of affective USV measurement, we inhibit neurons there using designer receptors exclusively activated by designer drugs (DREADDs). DREADDs are engineered G-protein-coupled receptors which do not respond to endogenous neurotransmitters and are thus inert—unless recruited by the experimenter via systemic application of the DREADD agonist clozapine n-oxide (CNO). In this way, DREADDs allow experimental “remote control” of neural populations during behavior [42,43,44]. We used a viral vector to express inhibitory DREADDs in NAc neurons under a synapsin promoter. This allows recruitment of inhibitory Gi/o-coupled signaling cascades in neurons, selectively inhibiting their activity without affecting glia, as could occur with conventional reversible inactivation approaches such as GABA receptor agonist-induced inhibition. We here asked whether NAc neuron DREADD inhibition impacts production of amphetamine-induced USVs. 

We found that chemogenetic suppression of NAc neural activity with DREADDs suppresses amphetamine-induced trill and non-trill frequency-modulated, high-frequency USVs. NAc inhibition did not affect high-frequency flat calls that were also increased by amphetamine, and did not induce low-frequency calls, which were rarely observed. These findings demonstrate that NAc is essential for some but not all amphetamine-induced USVs, showing that NAc neuron activity mediates amphetamine-induced subjective states in rats. 

## 2. Materials and Methods

All procedures were approved by the Institutional Animal Care and Use Committee at UC Irvine, and are in accordance with the NIH Guide for the Care and Use of Animals [45].

Subjects: Long Evans rats (*n* = 8 males, *n* = 8 females) were bred in-house, and housed as adults in pairs in ventilated tub cages with corncob bedding and ad libitum chow and water. Rats were at least 75 days old at the start of experiments. Rats were housed in reverse 12:12 h lighting, and behavior experiments took place during the dark cycle.

Drugs: CNO was graciously provided by the NIDA Drug Supply Program, stored in desiccated, opaque powder aliquots at 4 °C, and prepared daily, mixed in 5% dimethyl sulfoxide (DMSO) in saline solution. D-amphetamine hemisulfate salt was attained from Sigma and mixed in saline at 2 mg/mL. 

Viral Vector and Surgery: A previously validated [46] AAV2 hSyn-hM4Di-eGFP vector (titer ≥ 7 × 10^12^ vg/mL) was attained from AddGene. Rats were anesthetized with ketamine (56.5 mg/kg) and xylazine (8.7 mg/kg), with saline and meloxicam (1.0 mg/kg) analgesic, then stereotaxically injected via glass pipette and Picospritzer with 0.4 µL of the vector bilaterally, aimed at the nucleus accumbens core (relative to bregma (mm): +1.45 AP, ±1.3 ML, −7.6 DV). Pipettes were left in place for 5 min to reduce spread prior to removal. Rats were allowed at least 25 days to recover following surgery before handling and habituation. Vector expression is shown in Figure 1. 

Testing Protocol: Rats were handled daily for a week prior to habituation to the testing chamber, which occurred 1 h/day over 2 consecutive days. On each of the two subsequent test days, they received counterbalanced i.p. injections of CNO (5 mg/kg) or its vehicle (veh; 5% DMSO in saline), then were placed in the testing chamber for recording of USVs. 30 min later, they were injected with amphetamine (2 mg/kg) and returned to the recording chamber for another hour. At least 48 h later, the second test was conducted under the same protocol with the alternative CNO/veh treatment. 

USV Recording Apparatus: Testing was done in one of four clear acrylic tub cages (25 cm × 46 cm × 20 cm), with paper fiber bedding and a wire cage top. These testing chambers were enclosed within a wood sound-attenuating box (120 cm × 60 cm × 60 cm), which contained four bays in which testing cages could be placed, separated from each other by wood walls. These sound attenuating boxes helped exclude unwanted electrical and other noise in the 20–100 kHz range from recordings. The top of the sound-attenuating box was comprised of clear acrylic with a 3 cm hole drilled in the center of each bay, allowing placement of a USV microphone 33 cm above tub cages above the testing chambers, aimed at the center of each of the 4 testing cages. Avisoft Bioacoustics ultrasonic sensitive microphones (model CM16/CMPA; frequency range: 10–200 kHz), receivers (model 816H; sampling rate 250 kHz; 16-bit resolution), and PC software (Version 3.4.4) were employed, and recordings were done on a laptop running Windows. 

Algorithm-Detected USV Analyses: DeepSqueak software ([33]; DeepSqueak 2.0 with MATLAB) was employed to detect USVs and identify the detailed characteristics of human-verified USVs from this dataset. Audio files were run though DeepSqueak’s All Short Call neural network, then a post-hoc denoiser trained on noise inherent to the experimental setup automatically excluded non-USVs from the dataset. Call statistics for accepted USVs (20–100 kHz) were calculated using the spectrotemporal contours output by the detection network. These statistics included each call’s (1) principal frequency (average frequency over the call), (2) change in frequency over the course of the call, (3) sinuosity (length of the path between the first and last points on the contour, divided by the Euclidean distance between the first and last points), and (4) duration. 

Hand-Labeled USV Analysis: USVs identified by Deepsqueak were verified by a trained observer, blind to experimental conditions, using a previously reported scheme [24]. The observer verified around 95% of DeepSqueak-identified calls as valid, with 4160 calls out of 77,454 rejected as noise. USVs were visualized in spectrograms, which allowed inspection of call frequency over time. Putative calls were greater than 2 ms and were confirmed by the observer listening to them on headphones (frequency was transduced into the audible range by playing calls at 0.05 speed for this confirmatory evaluation). Identified calls were manually categorized into 1 of 4 categories: (1) Low-Frequency (LF) calls, which were between 20–30 kHz, (2) Flat high-frequency calls, which were between 30–100 kHz in principal frequency, and which lacked visible frequency modulation, (3) Frequency-Modulated (FM) calls, which had visible frequency modulation without a trill component, and (4) Trills, which had the same definitional criteria as FMs, but were at least 8 ms and contained 2 cycles of rapid oscillating frequency. 

Analyses and Statistics: For analyses of NAc inhibition effects on the number of each USV type emitted, USVs were quantified in 10 min bins, both in the 30 min baseline period (after veh/CNO but before amphetamine), and in the 1 h post-amphetamine testing period, on both test days (veh or CNO; Figure 2). ANOVAs with within-subjects Drug (veh and CNO days) and Time (three 10 min baseline bins, or six 10 min post-amphetamine bins) factors, and a between-subjects Sex (female/male) factor, with Tukey post hoc tests and Greenhouse-Geisser correction for normality assumption violations when needed, were employed. 

Call parameters were analyzed in detail for USVs emitted in the 1 h post-amphetamine session, on the vehicle pre-treatment day. USVs were analyzed for each of 4 measured call parameters (principal frequency, change in frequency, sinuosity, and duration). Averages for each rat on each parameter, categorized based on manually assigned USV types (LF, Flat, FM, Trill), were calculated, and group averages are shown in Figure 3A. For each measured parameter, call types were compared to one another using one-way ANOVA, with USV type treated as a between-subjects variable, since not all call types were emitted by all rats. This analysis served to confirm that Deepsqueak-derived USV parameters were logically related to human-assigned call categories, i.e., LF calls had low principal frequency, FMs and Trill had greater change in frequency than Flats, etc. To explore the underlying structure of amphetamine-induced USVs (on vehicle treatment day), we used Uniform Manifold Approximation and Projection (UMAP; [47]) to faithfully represent all four acoustic parameters derived by Deepsqueak in two dimensions, with results shown in Figure 3C. To determine NAc inhibition effects on characteristics of FM and Trill calls (those suppressed by CNO), for each parameter, t-tests comparing per-rat averages for USV parameters on vehicle or CNO amphetamine tests were used (Figure 2). 

## 3. Results

### 3.1. Localization of DREADD Expression

Each brain was examined for expression of mCherry, the DREADD tag in the employed vector (Figure 1A). In most brains, expression was largely localized to the nucleus accumbens core and shell, with some expression also seen in nearby regions such as the ventral caudate putamen, and lateral and medial septum, as shown in Figure 1B. 

### 3.2. Amphetamine Increases the Number of High Frequency USVs Emitted in Both Sexes

On vehicle day, amphetamine induced increases in all types of USVs except LFs, as determined by t-test comparing the 10 min prior to amphetamine (20–30 min after veh) to the first 10 min after amphetamine on that day (LF: t_15_ = 0.1; Flat: t_15_ = 3.5, *p* = 0.003; FM: t_15_ = 7.86, *p* < 0.001; Trill: t_15_ = 4.92, *p* < 0.001). The only major sex difference seen after amphetamine was that females produced more Trill calls than males (main effect of Sex: F_1,14_ = 5.5, *p* = 0.034). No effects of Sex on other call types, or interactions of Sex with pre versus post-amphetamine epochs were observed. Characteristic examples of each call type are shown in Figure 3B.

### 3.3. CNO Effects on the Number of USVs Emitted Prior to Amphetamine

To examine whether inhibition of NAc suppressed spontaneously emitted USVs even prior to amphetamine, we examined USVs that occurred in the 30 min after veh/CNO injection, during which period CNO effects were expected to emerge. Indeed, CNO suppressed certain USVs, but only late in the baseline period after it had time to reach the brain, as shown with interactions between Drug (veh/CNO) and Time (three 10 min bins) seen on FM (F_2,28_ = 7.0, *p* = 0.003) and Trill (F_2,28_ = 6.52, *p* = 0.005), but not LF (F_2,18_._2_ = 1.2, *p* = 0.31) or Flat USVs (F_2,28_ = 1.45, *p* = 0.25; Figure 2A,B). Main effects of Time also indicated that most calls declined over the baseline period independent of veh/CNO treatment (LF: F_2,28_ = 8.61, *p* = 0.001; Flat: F_2,28_ = 3.03, *p* = 0.06; FM: F_2,28_ = 5.89, *p* = 0.007; Trill: F_2,28_ = 3.82, *p* = 0.034). No significant main effects of Sex (ps > 0.075), nor interactions of Sex with other variables (ps > 0.2) were seen for any call type during baseline. This suggests that NAc inhibition modestly suppressed even the low levels of FM and Trill USVs seen prior to amphetamine administration. 

### 3.4. CNO Effects on the Number of Amphetamine-Induced USVs

NAc inhibition with DREADDs strongly and selectively suppressed amphetamine-induced FM and Trill calls (main effect of Drug; FM: F_1,14_ = 5.38, *p* = 0.036; Trill: F_1,14_ = 7.11, *p* = 0.018), but not Flat (F_1,14_ = 2.47, *p* = 0.14) or LF calls (F_1,14_ = 2.78, *p* = 0.12; Figure 2C). Since FM and Trill calls were similarly suppressed by CNO, we also analyzed data with these call types combined, and again found a robust main effect of Drug (F_1,14_ = 7.9, *p* = 0.014) and Time (F_1_._5,20_._97_ = 13.46, *p* < 0.001), as well as a main effect of Sex, with females emitting more total FM + Trill vocalizations (F_1,14_ = 4.95, *p* = 0.043). However, no interactions of Sex with Drug or Time were seen (ps > 0.52), indicating that inhibiting NAc had similar effects in both sexes (Figure 2D). Flat, FM and Trill calls were all maximal in the first 20 min after amphetamine, and decreased subsequently across the next 40 min in a similar manner after both veh and CNO (Main effect of Time; Flat: F_1_._43,20_._1_ = 5.36, *p* = 0.021; FM: F_1_._8,25_._5_ = 14.0, *p* < 0.001; Trill: F_1_._5,20_._4_ = 10.68, *p* < 0.001; Figure 2A,B). This suggests that CNO’s effect was to suppress the number of FM/Trill USVs seen, without impacting the timecourse of the USV-promoting effects of amphetamine. LF calls were rare after amphetamine, were almost exclusively “short,” non-canonical “22 kHz alarm calls,” and did not show a similar time course of amphetamine-induced increase to the other call types (F_1_._35,18_._91_ = 1.35, *p* = 0.25). A Wilcoxon rank sum test revealed that administration of amphetamine increased the proportion of Trills produced (W = 40, *p* < 0.001; Figure 2A), while reducing the proportion of both FM and Flat calls (FM: W = 212, *p* < 0.002; Flat: W = 187, *p* < 0.03). CNO administration did not change call subtype prevalence (ps > 0.05; Figure 2B).

### 3.5. Amphetamine-Induced USV Characteristics, and Sex Differences

Analysis of the acoustic characteristics of each of the four call types (examples shown in Figure 3B) revealed significant differences between amphetamine-induced USV types in each of the measured parameters (vehicle day; 1 h post-amphetamine period data were analyzed). USV types differed predictably (Figure 3A) in their principal frequency (F_3,51_ = 450.8, *p* < 0.0001) with LFs being lower-frequency than other call types (ps < 0.0001), and Trills having a higher principal frequency than the other calls (*p* < 0.0001). Also as expected, calls also differed in the degree to which frequency changes over the course of the call (F_3,51_ = 132.3 *p*< 0.0001), with Trills and FMs having a higher frequency variation than other calls (Trills: ps < 0.0001; FMs: ps < 0.0001), and Trills also being more modulated than FMs (*p* < 0.0001). Call types also differed in length (F_3,51_ = 5.42, *p* < 0.01), with Trills being longer than Flats, FMs, and LFs (ps < 0.03). For sinuosity, USV types also differed (F_3,51_ = 80.58, *p* < 0.0001), in that Trills had higher values than all other call types (ps < 0.0001), and FMs had higher values than Flats and LFs (ps < 0.0001). Collapsing across USV types, we saw no overt sex differences in the 4 measured call parameters, implying that amphetamine-induced USV characteristics are likely to be fundamentally similar regardless of sex (Figure 3D). In sum, though we are not convinced that Deepsqueak-derived USV parameters are precise in all cases, we nonetheless conclude based on these analyses that the software performs acceptably for roughly characterizing USV features under the recording conditions employed here. 

Therefore, we next conducted an exploratory, data-driven analysis examining how USV features relate to manually categorized USV subtypes and to sex (Figure 3C,E). Using UMAP [47,49,50,51,52], an advanced method of analyzing underlying features of large datasets such as this one, we plotted a projection of all 33,370 USVs quantified during the vehicle/amphetamine session in 4-dimensional space into 2 dimensions for easy visualization. In Figure 3C, each dot represents an emitted USV, color-coded based on manually assigned call type. Calls that are similar in all 4 dimensions cluster together, and groups of similar calls are represented in neighboring clusters. We found approximately 3 clusters of USV calls, each of which contained distinct proportions of each USV call type (Figure 3C). These data imply that Deepsqueak-quantified call parameters do not line up 1:1 with human-categorized USV categories, suggesting that these USV data contain an underlying architecture that is not captured by a simple 4 category quantification scheme, as employed here and in many other reports [24,25,26,53]. Finally, we took the same UMAP representation and color-coded USVs based on the sex of the rat that produced them (Figure 3E); we interpret results to show that both male and female rats show similar distribution of calls across all 3 clusters.

### 3.6. Effects of CNO on USV Characteristics

Inhibiting NAc did not affect the primary quantitative characteristics of FM and Trill USVs, though as described above, the number of these calls were suppressed (Figure 2A,B). Relative to vehicle, CNO failed to impact any of the parameters of FM (ts < 2.05; ps < 0.05), or Trill calls (ts < 1.49; ps < 0.15; Figure 4).

## 4. Discussion

To our knowledge, this is the first investigation on neural substrates of rodent ultrasonic vocalizations (USVs) employing a reversable chemogenetic inhibition approach. Results confirm that the nucleus accumbens (NAc) is a key locus in brain circuits underlying USVs emitted after systemic amphetamine injection. We show a preferential suppression of frequency-modulated vocalizations, both with and without trill components, after chemogenetic NAc neuronal inhibition with DREADDs. Although females emitted more FM/Trill USVs than males overall, DREADD inhibition suppressed these frequency-modulated, amphetamine-induced USVs to a similar degree in both sexes. Interestingly, non-frequency modulated high-frequency Flat USVs were induced mildly but significantly by amphetamine, but NAc inhibition did not alter their production, nor did it induce low-frequency, aversion-related USVs. Aversive low-frequency vocalizations are often hundreds of milliseconds long, and have been recorded after exposure to predator odor, foot shocks, and social isolation [14,15,16,18,19]. When we did see LF vocalizations here, they were much shorter, lasting less than 150 ms. These short LF calls, which are also seen during rat self-administration of cocaine and methamphetamine, may reflect less severe aversion than prototypical “22 kHz” alarm calls [24,25,26,54,55]. Regardless, amphetamine very rarely induced LFs of any duration, nor did NAc inhibition induce their production. Though NAc inhibition suppressed the quantity of emitted FM and Trill USVs, the quality of these calls as reflected in their quantitative parameters was not affected. Together, these data speak to the complexities of rat USVs both in their characteristics and their neural substrates, and show that detailed quantitative analyses of rat USVs could inform our understanding of the subjective effects of neural manipulations in preclinical behavioral neuroscience experiments. 

We show here that inhibition of NAc using an established chemogenetic inhibition approach [46,56,57,58] markedly suppresses amphetamine-induced USVs. Prior reports have shown that administration of amphetamine to the NAc shell increases local dopamine signaling and 50 kHz call rate [2,39,59,60], with similar increases in calling resulting from systemic or intracerebral injection of other dopamine agonists such as quinpirole [5,54,61,62,63]. Playback of 50 kHz vocalizations has also been shown to increase dopamine levels in the NAc and elicit approach behavior [41]. Disruption of dopamine signaling in the NAc through administration of dopamine antagonists or lesions to the ventral tegmental area suppresses 50 kHz calling [21,31,64,65,66]. Here, we used a viral vector causing expression of inhibitory DREADDs only in NAc neurons, without impacting activity of other NAc cell types which could be influenced by lesion or pharmacological inactivation approaches. When the DREADD agonist CNO is applied systemically, neurons are inhibited in their firing and neurotransmitter release [56,67]. We also found that NAc inhibition alone suppressed FM and Trill calls in a baseline period, prior to amphetamine administration. When amphetamine was injected, FM, Trill and Flat USVs increased markedly on both vehicle and CNO treatment days. Yet on the CNO day, FM and Trill calls (but not Flats) were suppressed relative to vehicle day, indicating that normal NAc neural activity is required for emission of these calls when either spontaneously emitted, or induced by a catecholamine-enhancing addictive drug. Further studies using more targeted chemo- or optogenetic manipulation approaches should explore further the mechanisms of this effect, such as the NAc neuron types, and the wider circuits interactions responsible for these effects. 

In addition to showing that NAc inhibition suppressed the number of USVs produced, we also sought to determine how quantitative analysis of USV parameters might inform our understanding of how these vocalizations could be used to interrogate the subjective states of rats, and the neural mechanisms thereof. Towards this aim, we employed a newly developed, machine learning-based software package for analyzing rat USVs, Deepsqueak [33]. We found Deepsqueak very useful for detecting USVs in our sound-attenuated setup. We did notice that some instances of noise were also registered as potential calls in our experiment, so we verified each auto-identified call using visual and audio inspection by a trained, blinded observer. This observer also manually categorized each call into one of 4 pre-defined categories, high frequency Flat, FM, and Trill calls, as well as aversion-related low frequency calls, based on prior work [24,26,68]. These pre-defined categories seemed to be supported by Deepsqueak-generated parameters, with logical characteristics generally captured (e.g. LF calls have lower frequency than high frequency calls, FM/Trill calls had more frequency modulation than Flat/LF calls, etc.). When we used these parameters to explore potential underlying relationships between emitted calls based on data structure rather than experimenter-determined categories, we found that our pre-set categories were imperfectly reflected in the data architecture, suggesting that hidden structures may exist in these data which is not completely captured by our literature-based USV categories. Moreover, we found little evidence for either sex or NAc inhibition affecting the quality of USVs produced. We strongly feel that further optimization of software solutions to accurately capture USV parameters, and examining them under a range of pharmacological, neural, and behavioral circumstances could hold the key to understanding the deeper meaning of rat USVs. 

Our study has a number of limitations: Sample size is relatively low (*n* = 8/sex), so although NAc inhibition effects were very clear, more subtle effects of sex or hormonal contributions may have been missed. Our viral injections were targeted at the NAc, although as is common in such experiments, not all DREADD expression was entirely restricted to the NAc, let alone NAc core or shell subregions. We also did not include a non-DREADD expressing group to control for potential off-target effects of CNO, which have been reported previously [69,70,71,72]. We therefore cannot exclude the possibility that CNO alone, even in the absence of DREADDs, could have impacted the results we observed. Additionally, we did not examine the impacts of NAc inhibition on saline-, rather than amphetamine-induced calling, which would have allowed better examination of chemogenetic impacts on non-pharmacologically-induced USV production. We thus focused our quantitative analyses of USVs on data generated after vehicle injection followed by amphetamine, since we were concerned that the relatively low number of USVs seen in the baseline period, and potential subtle effects of NAc inhibition on baseline or post-amphetamine responses could impact these analyses. Further work should involve recording of USVs under a variety of behavioral circumstances, neural states, and after dose-dependent pharmacological manipulations. 

In sum, these results confirm using a neuron-specific chemogenetic inactivation approach, that NAc is a key node in the neural circuits underlying rat USVs. Intriguingly, NAc seems to mediate frequency-modulated USVs in particular, but not high-frequency unmodulated calls that were also induced by amphetamine, nor the rarely observed, potentially aversion-related low-frequency USVs. Our results also point to the remaining mysteries surrounding these effect-relevant, but still poorly understood vocalizations, since the four pre-selected USV types did not clearly map onto clusters revealed by running UMAP on four quantitative USV parameters. We hope that with further refinement of USV recording and analysis methods, we may finally extract information about rat subjective states from their calls, which would be of tremendous benefit to preclinical behavioral neuroscience experiments. 

## Figures and Tables

**Figure 1 brainsci-11-01255-f001:**
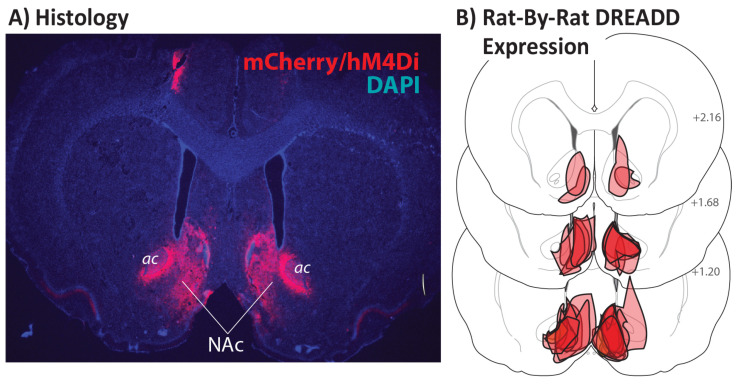
Nucleus Accumbens DREADD Expression: (**A**) Expression of the AAV2-hSyn-hM4Di-mCherry vector is depicted in a typical rat, shown in a coronal plane. mCherry and hM4Di are co-expressed in NAc neurons (red stain), and DAPI (4′,6-diamidino-2-phenylindole, in blue) defines anatomical landmarks, including anterior commissure (ac). (**B**) Viral expression localization is shown in each of the 16 tested animals in 3 coronal views of rat brain [48], with numbers indicating the approximate bregma-relative anterior coordinate.

**Figure 2 brainsci-11-01255-f002:**
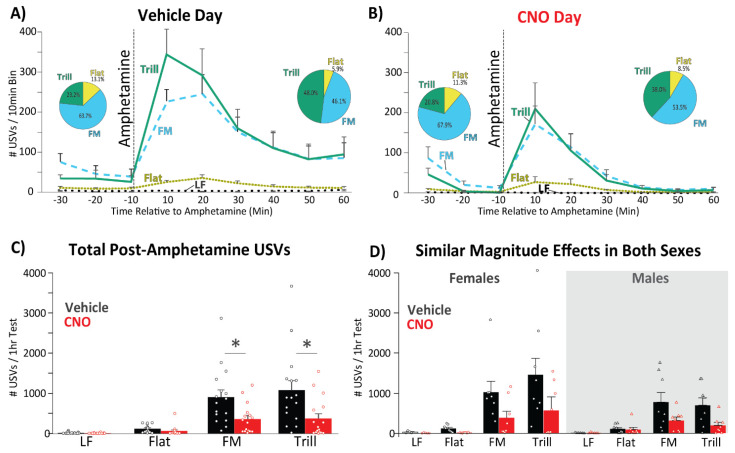
Nucleus Accumbens Inhibition Suppresses the Number of Frequency Modulated and Trill USVs: (**A**) USVs of each call type are depicted in 10 min bins in the post-vehicle, but pre-amphetamine baseline (left), and for 1 h after amphetamine (right; amphetamine injection occurred at the dotted vertical line). Pie charts show relative prevalence of each call subtype shown for pre-amphetamine baseline (left) and for 1 h after amphetamine (right). (**B**) CNO day data is shown in the same manner as in panel A. (**C**) Summary data are shown for the entire 1 h post-amphetamine session for the vehicle day (black) and CNO day (red), depicting main effects of this treatment. Dots depict data from individual animals. * *p* < 0.05. (**D**) Post-amphetamine summary data is shown in the same manner as in panel C, but separately in females (left) and males (right). Statistically similar CNO-suppression of the number of FM and Trill, but not Flat, USVs is seen in both sexes.

**Figure 3 brainsci-11-01255-f003:**
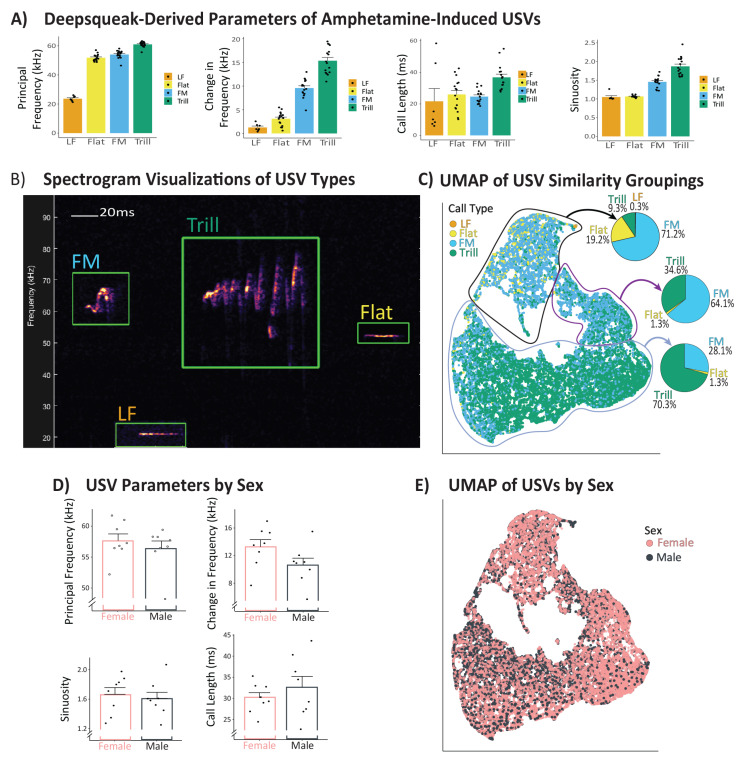
Characteristics of Quantified USVs: All USVs emitted on the vehicle + amphetamine test day were categorized by an observer into one of four categories: Low-frequency (LF), Flat, non-trill-containing frequency-modulated (FM), and trill-containing (Trill). Parameters (principal frequency, frequency modulation, duration, and sinuosity) for each emitted vocalization were quantified by Deepsqueak, and per-animal averages computed for each call type. (**A**) The mean and SEM of these per-rat average values for each call type are shown for each parameter, with dots indicating the mean value for each tested rat. (**B**) Examples of each of the 4 call types are shown. Scale bar indicates 20 ms. (**C**) UMAP representation of how calls relate to one another on the 4 measured parameters is depicted. Each emitted call is represented by a dot, which was color-coded based on manually assigned call type (LF = orange, Flat = yellow, FM = blue, Trill = green). Calls with similar characteristics in the 4 dimensions are represented within a cluster. To illustrate the proportion of call types present within visually determined clusters (circled), pie charts show the percentage of calls in each cluster that were assigned to each call type. Though each cluster differs in the types of calls contained within them, clusters do not neatly parse calls into our 4 pre-defined categories, which were based on the literature. This suggests that the 4-category system we used to define calls into subcategories may not fully capture underlying call subtypes present in the data, which could have functional significance. (**D**) Per-rat averages on each USV parameter, collapsed across call types, is shown broken down by the emitting animal’s sex. No significant sex differences were found, suggesting that rats do not differ based on sex in the fundamental features of emitted USVs. (**E**) Further demonstrating this point, the same UMAP plot is shown, with calls color-coded by sex rather than call type, demonstrating that both sexes emit calls throughout the entirety of 4-dimensional space, and in each of the data-derived clusters depicted.

**Figure 4 brainsci-11-01255-f004:**
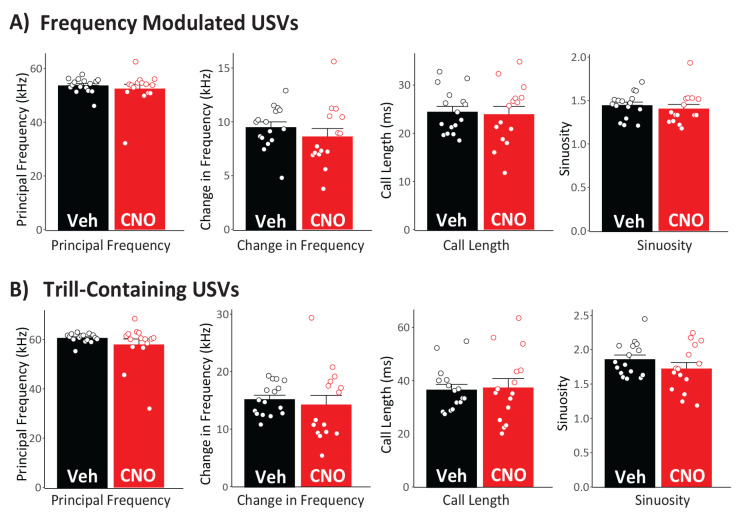
Nucleus Accumbens Inhibition Does Not Affect the Characteristics of Amphetamine-Induced USVs: Relative to vehicle day (black), USVs on CNO day (red) did not differ in principal frequency, frequency modulation, duration, or sinuosity for either (**A**) frequency-modulated non-trill, or (**B**) trill-containing USVs. Average values for each rat are shown with dots within bars.

## Data Availability

Processed data and code are available on the Mahler Lab GitHub (https://github.com/Mahler-Lab/NAc_USV accessed on 17 September 2021). Data will be made available upon request.

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
