# Peer review of "Nucleus Accumbens Chemogenetic Inhibition Suppresses Amphetamine-Induced Ultrasonic Vocalizations in Male and Female Rats"

_brainsci, 2021, doi:10.3390/brainsci11101255_

Round 1

Reviewer 1 Report

Lawson and colleagues studied amphetamine-induced ultrasonic vocalizations in rats and asked whether chemogenetic inhibition of the nucleus accumbens (using designer receptors exclusively activated by designer drugs; DREADDs) leads to reduced call emission. As expected, amphetamine (2 mg/kg) led to a prominent increase in the emission of high-frequency ultrasonic vocalizations, typically referred to as 50-kHz ultrasonic vocalizations. This increase was less prominent following chemogenetic inhibition of the nucleus accumbens. A moderate reduction in call emission was also seen before amphetamine treatment, suggesting that spontaneous ultrasonic calling is also affected by chemogenetic inhibition of the nucleus accumbens. Together, the study shows that the nucleus accumbens plays an important role in regulating the emission of 50-kHz ultrasonic vocalizations.

I very much enjoyed reading the manuscript. The manuscript is well written, the study well performed, and most of the conclusions well supported by the data. The extensive feedback that I provide below does therefore not primarily reflect the few weaknesses of the study but are motivated by my conviction that this is a very important study with high potential impact. It is a great study.

In my view, the description of the results is, at least in part, overly complex. Example: “We found that chemogenetic suppression of NAc neural activity with DREADDs suppresses amphetamine-induced trill and non-trill frequency modulated, high frequency USVs, without affecting high frequency flat calls that were also increased by amphetamine, or low frequency calls.” Looking at the findings, my conclusion would simply be: chemogenetic inhibition of the nucleus accumbens suppresses positive 50-kHz ultrasonic vocalizations. Aversive 22-kHz ultrasonic vocalizations were not observed. In fact, I would strongly recommend to start off the results sections with these two main types of ultrasonic vocalizations: the section on amphetamine effects with a brief description on how amphetamine administration affected positive 50-kHz ultrasonic vocalizations (but not aversive 22-kHz ultrasonic vocalizations, which did not occur) and the section on chemogentic inhibition with a brief description of how inhibiting the nucleus accumbens affected positive 50-kHz ultrasonic vocalizations (but not aversive 22-kHz ultrasonic vocalizations, which again did not occur). From there, one would could dig deeper and ask what specific 50-kHz call subtypes are primarily increased by amphetamine and which ones are inhibited by chemogenetic inhibition of the nucleus accumbens. The fact that the authors jump immediately into the complex call subtype analyses (and the fact that the most commonly used terms 22-kHz and 50-kHz ultrasonic vocalizations are not used – avoided?) makes it very difficult to understand the “big picture” and to link it back to the literature, particularly for the not so experienced readers.   

It is unfortunate that locomotor activity was not assessed. It would have been interesting to see whether amphetamine-induced hyperactivity is blocked by chemogenetic inhibition of the nucleus accumbens as well. Related to that, it is a bit unfortunate that amphetamine was not compared to vehicle. It was shown before that the amphetamine-induced increase in 50-kHz ultrasonic vocalizations is driven by frequency-modulated subtypes, including trills. Interestingly, these appear to be exactly the ones that are inhibited by chemogenetic inhibition of the nucleus accumbens. A saline control group would have allowed to test whether the amphetamine-induced shift in the call profile, i.e. subtype prevalence, is reversed by chemogenetic inhibition of the nucleus accumbens.

It is not really clear to me what data sets were analyzed using the software “deepsqueak” and which data sets were hand scored. Do I understand it right that all data sets were analyzed by “deepsqueak”  a n d  an experienced observer? If so, one could use these data sets to correlate call numbers between “deepsqueak” and hand scoring. One could also quantify the number of “false alarms” and “misses”. Moreover, one could ask whether the “false alarm” rate and the number of misses depends on the call subtype. All very interesting, in my view, because this would help to assess the validity of the “deepsqueak” approach.

As pointed out by the authors, a large area was affected by DREADD expression and expression is not limited to the nucleus accumbens. In general, variability is high. I thus wonder whether the size and/ or the pattern of the expression is associated with the strength of the inhibitory effect.

In the manuscript, the term “aversion-related LF calls” is used. I do not think that there is sufficient evidence supporting the notion that the observed low-frequency calls are related to aversion. The “LF calls” observed in the study were very short – and clearly shorter than typical 22-kHz ultrasonic vocalizations that have been linked to aversion. Such typical 22-kHz ultrasonic vocalizations last several hundred milliseconds up to 3-4 second. The “LF calls” observed in the study, however, were only about 20 milliseconds. Clearly, they do not belong to the group of 22-kHz ultrasonic vocalizations linked to aversion.

In the manuscript, it was said that all subtypes were collapsed for certain comparisons (e.g. line 204). Given the fact that low-frequency calls do not belong to the group of 50-kHz ultrasonic vocalizations and are not affected by amphetamine treatment, I would suggest to restrict this approach to subtypes that belong to the group of 50-kHz ultrasonic vocalizations. Of note, I am not so sure whether the low-frequency calls are actually ultrasonic vocalizations. Typically, a threshold of 20 kHz is used, albeit this threshold is artificial and based on our human hearing abilities (rats most likely do not care about), I feel like it makes sense to stick to this definition (for the sake of transparency and comparability). As shown in Figure 2A, some of the calls appear to be below that threshold.  

I find the UMAP approach very interesting but I am not surprised that there is no overlap in terms of UMAP clusters and call clusters obtained during hand scoring. Typically, clustering by a human is primarily based on three to four dimensions: principal frequency, frequency modulation, sinuosity, and, in a few cases, duration. Amplitude is typically not taken into account. The UMAP approach, in contrast, takes all these five parameters into account and it is therefore not surprising the result pattern differs. I therefore would suggest to compare the UMAP approach restricted to principal frequency, frequency modulation, and sinuosity with hand scoring. In fact, amplitude is not a very good parameter in general because the rats are moving around in the test environment and with the distance changing between rat and microphone the loudness changes. Therefore, this parameter is very “noisy”. Because of these reasons, I do not believe in the “hidden structure that may exist” (line 351). Of note, I do not understand why the power values are negative.

A few conclusions are not fully supported by the data. Example: “Yet on the CNO day, FM and Trill calls (but not Flats) were suppressed relative to vehicle day, indicating that normal NAc neural activity is required for emission of these calls when either spontaneously emitted, or induced by a catecholamine-enhancing addictive drug.” I do not think that “required” is the correct term here. If “normal NAc neural activity” would have been “required for emission of these calls”, no such call should have been emitted. What was seen was a moderate decrease rather than a complete blockade. Same applies to “necessary” (line 373). 

Suggested references:

  • Redecker et al., BBR, 2019, similarly performed a comparison of acoustic features between types of 50-kHz ultrasonic vocalizations. It would be interesting to compare the findings.
  • Wohr, BJP, 2021, recently reviewed the literature on amphetamine-induced 50-kHz ultrasonic vocalizations.

Minor points:

In the abstract, please introduce the abbreviation “USVs”.

In the introduction, references 31-34 do not fit very well. References 31-34 are not about advances in microphone technology or software developments for more accurate sound analysis. They rather deal with localizing the sender emitting ultrasonic vocalizations in a social context with several mice or rats.

In the results, the first letter of the term “sex” is often capitalized. Why?  

Author Response

Lawson and colleagues studied amphetamine-induced ultrasonic vocalizations in rats and asked whether chemogenetic inhibition of the nucleus accumbens (using designer receptors exclusively activated by designer drugs; DREADDs) leads to reduced call emission. As expected, amphetamine (2 mg/kg) led to a prominent increase in the emission of high-frequency ultrasonic vocalizations, typically referred to as 50-kHz ultrasonic vocalizations. This increase was less prominent following chemogenetic inhibition of the nucleus accumbens. A moderate reduction in call emission was also seen before amphetamine treatment, suggesting that spontaneous ultrasonic calling is also affected by chemogenetic inhibition of the nucleus accumbens. Together, the study shows that the nucleus accumbens plays an important role in regulating the emission of 50-kHz ultrasonic vocalizations.

I very much enjoyed reading the manuscript. The manuscript is well written, the study well performed, and most of the conclusions well supported by the data. The extensive feedback that I provide below does therefore not primarily reflect the few weaknesses of the study but are motivated by my conviction that this is a very important study with high potential impact. It is a great study.

We thank the reviewer for her/his interest and enthusiasm for the manuscript.

 In my view, the description of the results is, at least in part, overly complex. Example: “We found that chemogenetic suppression of NAc neural activity with DREADDs suppresses amphetamine-induced trill and non-trill frequency modulated, high frequency USVs, without affecting high frequency flat calls that were also increased by amphetamine, or low frequency calls.” Looking at the findings, my conclusion would simply be: chemogenetic inhibition of the nucleus accumbens suppresses positive 50-kHz ultrasonic vocalizations.

We appreciate this point, and have taken steps to streamline the results and interpretations in response. For example, we now make clear in several places that low frequency calls were rarely seen, and when they were seen they differed from longer “alarm” or “22 kHz” calls (e.g. lines 315-323). However, we also note that NAc DREADD inhibition suppressed the number of FM and Trill calls, but did not suppress the (admittedly lower) number of flat, high-frequency calls, which were also induced by amphetamine. We find this distinction interesting, in that only frequency modulated (with or without Trill), amphetamine-induced calls seem to be affected by NAc inhibition. We propose this could be important in the ongoing effort by many labs to map USVs onto specific brain circuits, in pursuit of better understanding their functions and significance.

Aversive 22-kHz ultrasonic vocalizations were not observed. In fact, I would strongly recommend to start off the results sections with these two main types of ultrasonic vocalizations: the section on amphetamine effects with a brief description on how amphetamine administration affected positive 50-kHz ultrasonic vocalizations (but not aversive 22-kHz ultrasonic vocalizations, which did not occur) and the section on chemogenetic inhibition with a brief description of how inhibiting the nucleus accumbens affected positive 50-kHz ultrasonic vocalizations (but not aversive 22-kHz ultrasonic vocalizations, which again did not occur).

We have adopted these suggestions, and believe the suggested organization has improved the manuscript.

From there, one would could dig deeper and ask what specific 50-kHz call subtypes are primarily increased by amphetamine and which ones are inhibited by chemogenetic inhibition of the nucleus accumbens. The fact that the authors jump immediately into the complex call subtype analyses (and the fact that the most commonly used terms 22-kHz and 50-kHz ultrasonic vocalizations are not used – avoided?) makes it very difficult to understand the “big picture” and to link it back to the literature, particularly for the not so experienced readers.   

We have incorporated this excellent suggestion as well. We now organize results in a more straightforward manner, and have added an additional analysis to examine the ratios of call types emitted in pre-amphetamine and post-amphetamine periods on vehicle and CNO days. These results are now presented statistically on lines 235-239, and also graphically in pie charts added to Figure 2A&B.

It is unfortunate that locomotor activity was not assessed. It would have been interesting to see whether amphetamine-induced hyperactivity is blocked by chemogenetic inhibition of the nucleus accumbens as well. Related to that, it is a bit unfortunate that amphetamine was not compared to vehicle. It was shown before that the amphetamine-induced increase in 50-kHz ultrasonic vocalizations is driven by frequency-modulated subtypes, including trills. Interestingly, these appear to be exactly the ones that are inhibited by chemogenetic inhibition of the nucleus accumbens. A saline control group would have allowed to test whether the amphetamine-induced shift in the call profile, i.e. subtype prevalence, is reversed by chemogenetic inhibition of the nucleus accumbens.

We agree that these omissions, necessary due to funding considerations, are unfortunate. We now acknowledge the lack of a saline control group (also mentioned by reviewer 2) on p.11. “Additionally, we did not examine impacts of NAc inhibition on saline-, rather than amphetamine-induced calling, which would have allowed better examination of chemogenetic impacts on non-pharmacologically induced USV production”. However, toward the reviewer’s broader point we now show the proportion of calls emitted pre/post amphetamine on vehicle and CNO tests days in Figure 2A&B, which we believe is a useful addition.

It is not really clear to me what data sets were analyzed using the software “deepsqueak” and which data sets were hand scored. Do I understand it right that all data sets were analyzed by “deepsqueak”  and  an experienced observer? If so, one could use these data sets to correlate call numbers between “deepsqueak” and hand scoring. One could also quantify the number of “false alarms” and “misses”. Moreover, one could ask whether the “false alarm” rate and the number of misses depends on the call subtype. All very interesting, in my view, because this would help to assess the validity of the “deepsqueak” approach.

We have clarified our precise scoring methods as suggested, and now clearly state that Deepsqueak was used to identify potential calls, which were each verified as genuine, non-noise USVs using visual and auditory evidence. We now estimate our “false alarm” rate from this method on p.3 line 137-9: “The observer verified around 95% of DeepSqueak-identified calls as valid, with 4,160 calls out of 77,454 rejected as noise.”

As pointed out by the authors, a large area was affected by DREADD expression and expression is not limited to the nucleus accumbens. In general, variability is high. I thus wonder whether the size and/ or the pattern of the expression is associated with the strength of the inhibitory effect.

Unfortunately, due to the variable nature of the precise patterns of extra-accumbens viral expression spread across animals, as well as the well-known variability in USV production we also saw, we were insufficiently powered to conduct this structure-function relationship analysis with this dataset.

In the manuscript, the term “aversion-related LF calls” is used. I do not think that there is sufficient evidence supporting the notion that the observed low-frequency calls are related to aversion. The “LF calls” observed in the study were very short – and clearly shorter than typical 22-kHz ultrasonic vocalizations that have been linked to aversion. Such typical 22-kHz ultrasonic vocalizations last several hundred milliseconds up to 3-4 second. The “LF calls” observed in the study, however, were only about 20 milliseconds. Clearly, they do not belong to the group of 22-kHz ultrasonic vocalizations linked to aversion.

This is a fair point, and we have adjusted our language throughout the manuscript to address it. In particular, we now discuss the distinction between “short” low frequency calls (occasionally seen here), and “typical” longer ones not seen here on lines 315-323:

“Aversive low frequency vocalizations are often hundreds of milliseconds long, and have been recorded after exposure to predator odor, foot shocks, and social isolation [14-16,18,19]. When we did see LF vocalizations here, they were much shorter, lasting less than 150 ms. These short LF calls, which are also seen during rat self-administration of cocaine and methamphetamine, may reflect less severe aversion than prototypical “22 kHz” alarm calls [24-26,54,55]. Regardless, amphetamine very rarely induced LFs of any duration, nor did NAc inhibition induce their production.”

In the manuscript, it was said that all subtypes were collapsed for certain comparisons (e.g. line 204). Given the fact that low-frequency calls do not belong to the group of 50-kHz ultrasonic vocalizations and are not affected by amphetamine treatment, I would suggest to restrict this approach to subtypes that belong to the group of 50-kHz ultrasonic vocalizations. Of note, I am not so sure whether the low-frequency calls are actually ultrasonic vocalizations. Typically, a threshold of 20 kHz is used, albeit this threshold is artificial and based on our human hearing abilities (rats most likely do not care about), I feel like it makes sense to stick to this definition (for the sake of transparency and comparability). As shown in Figure 2A, some of the calls appear to be below that threshold.  

We agree with the reviewer that there were issues with out prior data on the (few) low frequency vocalizations we observed. Some of these vocalizations, though validated by a human observer as veridical, contained noise components that impacted the ability of Deepsqueak to correctly compute parameters for them. We therefore now take a more conservative approach as requested, and exclude all calls with a calculated principal frequency <20 kHZ from the dataset (n=602 omitted from the prior submission). 87 low frequency calls now remain, and since we are confident in their characterization, we present these descriptive data in Figure 3. 

I find the UMAP approach very interesting but I am not surprised that there is no overlap in terms of UMAP clusters and call clusters obtained during hand scoring. Typically, clustering by a human is primarily based on three to four dimensions: principal frequency, frequency modulation, sinuosity, and, in a few cases, duration. Amplitude is typically not taken into account. The UMAP approach, in contrast, takes all these five parameters into account and it is therefore not surprising the result pattern differs. I therefore would suggest to compare the UMAP approach restricted to principal frequency, frequency modulation, and sinuosity with hand scoring.

We were also surprised that our UMAP cluster structure did not better capture the pre-hoc categories we manually assigned. We took the reviewer’s excellent suggestion to omit Power from the analysis, since as s/he points out, this is likely highly impacted by animal position or other irrelevant factors. However, when re-conducting UMAPs using only principal frequency, change in frequency, sinuosity, and duration, we still do not see the clusters neatly lining up with human categorization. In particular, Trills and other FM calls frequently cluster together in a manner that suggests that there could be subtypes of each that relate more to each other than to others within their manually-assigned category. There could be a variety of reasons for this outcome, but we are most intrigued by the possibility that this could reflect “hidden structure” within the data that transcends the pre-defined categories, as we mention on lines 365-9. Though tentative, we hope this proposition will spur others to take similar data-driven approaches to uncover any such hidden meaning within large USV datasets that could shed light on the subjective states of animals.      

In fact, amplitude is not a very good parameter in general because the rats are moving around in the test environment and with the distance changing between rat and microphone the loudness changes. Therefore, this parameter is very “noisy”. Because of these reasons, I do not believe in the “hidden structure that may exist” (line 351). Of note, I do not understand why the power values are negative.

As mentioned above, we agree, and have removed power as a variable in the paper.

A few conclusions are not fully supported by the data. Example: “Yet on the CNO day, FM and Trill calls (but not Flats) were suppressed relative to vehicle day, indicating that normal NAc neural activity is required for emission of these calls when either spontaneously emitted, or induced by a catecholamine-enhancing addictive drug.” I do not think that “required” is the correct term here. If “normal NAc neural activity” would have been “required for emission of these calls”, no such call should have been emitted.

We have adjusted our language describing this effect as suggested.

What was seen was a moderate decrease rather than a complete blockade. Same applies to “necessary” (line 373). 

We have also adjusted our language here.

Suggested references:

  • Redecker et al., BBR, 2019, similarly performed a comparison of acoustic features between types of 50-kHz ultrasonic vocalizations. It would be interesting to compare the findings.
  • Wohr, BJP, 2021, recently reviewed the literature on amphetamine-induced 50-kHz ultrasonic vocalizations.

 We have added these references.

Minor points:

In the abstract, please introduce the abbreviation “USVs”.

Done

In the introduction, references 31-34 do not fit very well. References 31-34 are not about advances in microphone technology or software developments for more accurate sound analysis. They rather deal with localizing the sender emitting ultrasonic vocalizations in a social context with several mice or rats.

We have edited these references accordingly.

In the results, the first letter of the term “sex” is often capitalized. Why?  

We intended capitalization to reflect use of Sex as a factor in ANOVA analyses, but have removed a few cases in which capitalization was inadvertently used otherwise. We thank the reviewer for drawing our attention to this error.

Reviewer 2 Report

The authors studied the effects of NAc inhibition on amphetamine-induced USV calls in rats. Overall, the manuscript is well organized and the findings are potentially interesting. I only have a few minor comments.

  1. One important saline control is missing. It is unclear at this point whether the amphetamine-induced USV calls are specific to amphetamine, rather than i.p injections in general.
  2. It is unclear to me what the principal frequency is referring to. Figure 3 shows CNO reduces the number of calls, doesn't it refer to a decrease in frequency in a given time?
  3. Is the UMAP embedding generated by a customized code? If so, please provide an open-source repository for the code.

Author Response

The authors studied the effects of NAc inhibition on amphetamine-induced USV calls in rats. Overall, the manuscript is well organized and the findings are potentially interesting. I only have a few minor comments.

1. One important saline control is missing. It is unclear at this point whether the amphetamine-induced USV calls are specific to amphetamine, rather than i.p injections in general.

As mentioned above in response to this fair point (also brought up by reviewer 1), we now explicitly state this as a limitation of the study on p. 11: “Additionally, we did not examine impacts of NAc inhibition on saline-, rather than amphetamine-induced calling, which would have allowed better examination of chemogenetic impacts on non-pharmacologically induced USV production.”

2. It is unclear to me what the principal frequency is referring to. Figure 3 shows CNO reduces the number of calls, doesn't it refer to a decrease in frequency in a given time?

We apologize for the confusion, but believe it stems from the two meanings of “frequency” that are used here and elsewhere in this literature. The reviewer is referring to the rate of calling (i.e. number of calls over a period of time), which as he or she points out is reduced by CNO, shown in Figure 2 (previously Figure 3).

“Principal frequency,” or the mean frequency of sound over the duration of a call (in kHz), is depicted in Figures 3&4. The principal frequency of amphetamine-induced Trills and other FM calls is not apparently affected by CNO (Figure 4). 

3. Is the UMAP embedding generated by a customized code? If so, please provide an open-source repository for the code.

We thank the reviewer for this excellent suggestion, and now provide a link to the code on p.11 in the data availability statement.

Round 2

Reviewer 1 Report

The authors thoroughly revised the manuscript and addressed all my comments. I have only one minor comment. Similar to lines 41-42 ("Lower frequency USVs 41 (LF; 18-30kHz; also called “22kHz” vocalizations) are..."), I suggest adding "also called "50kHz" vocalizations" to one of the next sentences, e.g. line 47. This probably sounds like a very minor point, but it is not. The terms "22-kHz USV" and "50-kHz USV" are used by the vast majority of publications in the field. Avoiding such terms makes it a lot more difficult to find this important study - which would be unfortunate. For this reason (i.e. to allow search engines to find the study), I suggest including these terms as key words as well. (Many of the key words are used in the title and could be replaced.) 

Author Response

We have added "50 kHz vocalizations" and "22 kHz vocalizations to keywords, and added "22 kHz vocalizations" as requested on line 47.

We thank the reviewer and editor again for improving the submission with their thorough reviews.